# Establishing Irreversible Electroporation Electric Field Potential Threshold in A Suspension In Vitro Model for Cardiac and Neuronal Cells

**DOI:** 10.3390/jcm10225443

**Published:** 2021-11-22

**Authors:** Sahar Avazzadeh, Barry O’Brien, Ken Coffey, Martin O’Halloran, David Keane, Leo R. Quinlan

**Affiliations:** 1Physiology and Cellular Physiology Research Laboratory, School of Medicine, Human Biology Building, National University of Ireland, H91 TK33 Galway, Ireland; sahar.avazzadeh@nuigalway.ie; 2AtriAN Medical Limited, Unit 204, NUIG Business Innovation Centre, Upper Newcastle, H91 TK33 Galway, Ireland; barry.obrien@atrianmedical.com (B.O.); ken.coffey@atrianmedical.com (K.C.); 3Translational Medical Devise Lab, Lambe Institute of Translational Research, University College Hospital Galway, H91 TK33 Galway, Ireland; martin.ohalloran@nuigalway.ie; 4Electrical & Electronic Engineering, School of Engineering, National University of Ireland Galway, H91 TK33 Galway, Ireland; 5Cardiac Arrhythmia Service, St Vincent’s University Hospital, D04 T6F4 Dublin, Ireland; dkeane@svhg.ie; 6CÚRAM SFI Centre for Research in Medical Devices, National University of Ireland, H91 TK33 Galway, Ireland

**Keywords:** atrial fibrillation, cardiac ablation, irreversible electroporation

## Abstract

Aims: Irreversible electroporation is an ablation technique being adapted for the treatment of atrial fibrillation. Currently, there are many differences reported in the in vitro and pre-clinical literature for the effective voltage threshold for ablation. The aim of this study is a direct comparison of different cell types within the cardiovascular system and identification of optimal voltage thresholds for selective cell ablation. Methods: Monophasic voltage pulses were delivered in a cuvette suspension model. Cell viability and live–dead measurements of three different neuronal lines, cardiomyocytes, and cardiac fibroblasts were assessed under different voltage conditions. The immediate effects of voltage and the evolution of cell death was measured at three different time points post ablation. Results: All neuronal and atrial cardiomyocyte lines showed cell viability of less than 20% at an electric field of 1000 V/cm when at least 30 pulses were applied with no significant difference amongst them. In contrast, cardiac fibroblasts showed an optimal threshold at 1250 V/cm with a minimum of 50 pulses. Cell death overtime showed an immediate or delayed cell death with a proportion of cell membranes re-sealing after three hours but no significant difference was observed between treatments after 24 h. Conclusions: The present data suggest that understanding the optimal threshold of irreversible electroporation is vital for achieving a safe ablation modality without any side-effect in nearby cells. Moreover, the evolution of cell death post electroporation is key to obtaining a full understanding of the effects of IRE and selection of an optimal ablation threshold.

## 1. Introduction

The therapeutic value of cardiac ablation has been transformed through iterative technology development, leading to enhanced treatment safety and efficacy for atrial fibrillation (AF), atrial flutter, and ventricular arrhythmias. AF is the most common form of cardiac arrhythmia widely associated with increased age, though also occurs in young adults and adolescent [1]. In younger adults, AF is usually precipitated by many underlying factors such as hypertension, hyperthyroidism [2], alcohol consumption, smoking [3], and channelopathies [4].

Cardiac ablation aims to destroy arrhythmogenic tissue, creating a permanent lesion. This approach initially started with using energy sources such as direct current, and radiofrequency. While radiofrequency and cryothermal ablation are largely efficacious and continue to be a mainstay of the current therapeutic arsenal, there is a need for an alternative ablation strategy improving cell targeting and ablation safety [5]. More recently, irreversible electroporation (IRE) has been proven to be a minimally thermal, safe, and effective technique for ablating a range of tissues [6,7]. For cardiac applications in particular, IRE offers a number of advantages compared to thermal-based methods, reducing treatment time and mitigating the risks of collateral damage [8,9]. IRE for cardiac ablation was first reported in vivo, which suggested that selective ablation for various tissue types was possible with this approach [10]. The IMPULSE First-in-Human clinical trial performed pulmonary vein isolation in patients with paroxysmal atrial fibrillation where the preferential ablation of myocardial tissue compared to collateral structures was clearly demonstrated [11]. However, it is challenging to interpret these results considering the in vivo 3D geometry of tissues, their proximity to the electrode, and local electric field strength. It remains unclear which level of electric field, pulse number, and pulse duration are optimal to obtain tissue-specific cell death.

At a cellular level, IRE results in the formation of hydrophilic nanopores in the cell membrane with permanent, long-lasting/irreversible effects on permeability across the phospholipid membrane [12]. IRE ablation of rat cardiomyocytes (H9C2) showed that field strengths greater than 375 V/cm significantly damaged cardiomyocytes [13]. Hunter D et al. reported biphasic pulses of 500 V/cm caused 80% cell death in a monolayer of rat ventricular cardiomyocytes [14]. There is the potential for significant advances in the ability to target cells in complex tissues with IRE through optimisation of therapeutic variables such as pulse duration, frequency, amplitude, and shape. Optimisation of these parameters must be based on solid experimental data if it is to yield advances in selective efficacy. There are very few reports to date reporting on comparative IRE thresholds for cardiac cells relative to other appropriate cardiac-neuronal model systems.

The aim of this study was to establish the IRE ablation threshold in a cell suspension model with cell types relevant to cardiac ablation. In addition, we investigated the temporal dynamics of RE and delayed IRE in these cells.

## 2. Methods

### 2.1. Cell Culture and Culture Conditions

PC12 cells (ATCC, pheochromocytoma cells derived from *Rattus norvegicus* adrenal glands), F11 (Sigma-Aldrich, somatic cell hybrid of rat embryonic dorsal root ganglion and mouse neuroblastoma cell line N18TG2), and SH-S5Y5 (Sigma-Aldrich, human neuroblastoma) were cultured in T75 flasks with DMEM (Sigma-Aldrich), supplemented with 10% foetal bovine serum (Sigma-Aldrich, Burlington, MA, USA) and 1% penicillin/streptomycin (10,000 U/mL, Gibco). HL-1 (Sigma-Aldrich, immortalised mouse atrial cardiomyocytes) were cultured in T75 in claycomb medium supplemented with 10% HL-1 foetal bovine serum, 1% penicillin/streptomycin, 2 mM L-glutamine, and 0.1 mM norepinephrine (Sigma-Aldrich). Human cardiac fibroblasts (Sigma-Aldrich, ventricular of adult heart) were grown in specialised cardiac fibroblast medium (Sigma-Aldrich). All cells were cultured at 37 °C in a humidified environment containing 5% CO_2_ and sub-cultured with trypsin-EDTA 0.025% (Sigma-Aldrich).

### 2.2. Electric Field Generation

Cells in culture at 70–90% confluence were detached and re-suspended in pre-warmed phosphate buffered saline (PBS) in cell viability assay or DMEM at live-death assay at concentration of 4–4.5 × 10^5^ cells/mL and transferred to a 4 mm gap electroporation cuvette (BTX, Harvard Apparatus, Figure 1A). Suspended cells were exposed to a voltage protocol (100 μs, monopolar pulses, inter pulse interval of 1 second, Figure 1B) of different field strengths (12.5, 200, 500, 1000, and 1250 V/cm) and pulse numbers controlled by a commercial pulse generator (BTX Gemini, Harvard Apparatus).

### 2.3. Cell Viability Assay

Immediately after electroporation, the cell suspension was transferred from the cuvette to a 48-well plate and incubated with 10% of 0.15 mg/mL Resazurin (Alamar blue assay) for three hours at 37 °C. The florescence intensity was measured using the Hidex microplate reader (Hidex Sense) at excitation/emission of 560/590 nm.

### 2.4. Live–Dead Assay

Cells were electroporated in standard growth media and subsequently transferred to a 48-well plate and incubated at 37 °C. At different time points post electroporation 0.5, 3, and 24 h, propidium iodide (PI, Sigma-Aldrich) was added at a final concentration of 1.5 μM and the cells maintained at 37 °C for 40 min before analysis. Florescence intensity was measured using Hidex microplate reader (Hidex Sense) at excitation/emission of 520/620 nm. The fluorescent intensity of PI^+^ cells were normalised to control (non-treated) cells and expressed as a fold change.

### 2.5. Immunocytochemistry Staining

Cultured cells were washed with PBS and fixed in 4% paraformaldehyde, blocked for one hour with 0.2% bovine serum albumin (in 0.1% Trition-X100), and incubated with myosin 4 monoclonal antibody (ThermoFisher Scientific, Waltham, MA, USA) at 4 °C overnight. The following day the cells were washed and incubated with anti-mouse 488 fluorophore conjugated secondary antibody (Sigma, SAB4600387, 77671-1ML-F, 1:1000) in blocking solution for one hour at room temperature. Cells were washed three times in PBS and then imaged using the EVOS microscope system. In all cases, nuclei were counterstained with DAPI.

### 2.6. Statistical Analyses

All data were analysed using paired *t*-test or two-way ANOVA. All experiments repeated for at least three independent experimental blocks.

## 3. Results

### 3.1. Reduction in Cell Viability Is Related to Electric Field Strength

PC12, F11, and SH-S5Y5 cells (Figure 2A–C) in suspension culture were treated with different electric field strengths (12.5, 200, 500, 1000, 1250 V/cm) and pulse numbers (10, 30, 50, 60). In all lines tested, cell viability showed a significant reduction after exposure to fields greater than 1000 V/cm with 30 pulses or more (Figure 2D–F). The optimal threshold was defined as the minimum electric field and pulse number that resulted in an 80% reduction in cell viability. Based on these criteria, the threshold for all three neuronal lines was 1000 V/cm with 50 pulses (Figure 2D–F). There was no significant difference in cell viability between neuronal cell lines at this voltage level (Figure 2G,H). When considering HL-1 cardiomyocytes and cardiac fibroblasts (Figure 3A–C), the viability of HL-1 cardiomyocyte was significantly reduced by 60 pulses at a field of 500 V/cm (Figure 3D). However, based on our criteria, the lethal threshold was at higher voltages of 1000 and 1250 V/cm with at least 30 pulses (Figure 3D). In contrast, cardiac fibroblasts were significantly less sensitive to voltage compared to cardiomyocytes. At the higher field strengths of 1250 V/cm (50 and 60 pulses) and 1000 V/cm (60 pulses) cardiac fibroblasts showed 10% viability (Figure 3E). The direct comparison of the three highest field strengths employed in this study, 500 V/cm (Figure 3F), 1000 V/cm (Figure 3G) and 1250 V/cm (Figure 3H) shows a trend toward decreased viability with a significant difference at 30 and 50 pulses at 1000 V/cm (Figure 3G).

The lethal electroporation threshold of all neuronal lines versus cardiomyocytes and cardiac fibroblasts is not significantly different (Figure 4A–E). Based on our criteria, the lethal threshold for IRE is very similar for neurons and cardiac cells in the suspension culture model used in this study.

### 3.2. Effect of Time on Cell Death Post Electroporation

Cells were exposed to the lethal threshold established in earlier experiments and live–dead analysis was assessed with PI staining at 0.5, 3, and 24 h post electroporation. PC12 showed a large fold change in cells exhibiting permeability to PI at 0.5 h following electroporation. However, PI uptake was significantly reduced after 3 and 24 h at 30 pulses for all field strengths tested (Figure 5A). This suggests that a proportion of cells that are initially permeable to PI, overtime re-sealed when a lower number of pulses are applied. In contrast, with 60 pulses, there was a significant reduction in PI uptake after only three hours, whereas after 24 h, the percentage of PI^+^ cells returned to their initial level. These data suggest that after three hours, the membrane of some cells re-sealed, resulting in a drop in PI^+^ cells. However, it is also clear that a significant percentage of cells are PI^+^ at 24 h. This suggests that either a different cell death pathway is responsible for cell death observed after 24 h, or that with 60 pulses, the proportion of cells resealing and surviving is reduced.

For F11 cells, following electroporation, cells showed consistent and stable PI uptake at 24 h independent of treatment (Figure 5B). In contrast to the other neuronal lines, SH-S5Y5 cells showed significant increase after three and 24 h, suggesting activation of a delayed cell death pathway in these cells (Figure 5C).

The PI^+^ fold change difference between all neuronal lines remained consistent with the Alamar blue viability assay with no significant differences. Direct comparison of immediate cell death (30 min post IRE) in F11 showed a significant reduction versus PC12 cells at all field strengths and only at 1250 V/cm versus SH-S5Y5 cells (Table 1). PC12 cells had significantly more PI uptake versus both F11 and SH-S5Y5 cells.

Membrane permeability of HL-1 cardiomyocytes and cardiac fibroblasts to PI was also investigated at three different time points after IRE. Permeability of cardiomyocytes to PI with 30 pulses showed no significant difference across different time points for all field strengths, however, there was a significant delayed cell death with 60 pulses (Figure 6A). In contrast, cardiac fibroblasts had a significant increase in PI^+^ cells after 24 h, showing a trend toward increased cell death overtime (Figure 6B). The immediate PI^+^ fold change (0.5 h) was significantly higher in cardiomyocytes in comparison to cardiac fibroblasts at higher field strength of 1250 V/cm (Table 2).

## 4. Discussion

In this study, we used a simplified suspension culture electroporation model to explore IRE thresholds for cardiac applications. Our data showed that IRE produces significant cell death at field strengths greater than 1000 V/cm, when deployed with at least 50 pulses. The IRE lethal threshold was very similar across all neuronal lines despite their differences in tissue origin (Figure 2G,H). However, there was a significant difference in lethal ablation threshold when comparing cardiomyocytes and cardiac fibroblasts, with cardiomyocytes showing significantly higher cell death (Figure 3F–H). Cell permeability to PI was measured immediately after electroporation and the data showed a significant increase in PI uptake in neuronal lines (PC12 and SH-S5Y5) compared to HL-1 and cardiac fibroblasts. The differences in PI uptake diminished over time in PC12 cells with lower pulse numbers. However, this was not apparent at higher pulse numbers, with similar cell death between 0.5- and 24-h time points. Cardiomyocytes and cardiac fibroblasts showed a degree of delayed cell death with significant changes at 24 h, suggesting that a different cell death mechanism may be active in these cells. These data suggest that in terms of cell selectivity with IRE, the lethal threshold for neuronal and cardiac cells are very similar with the primary determinant being field strength.

We also established that while above the IRE threshold caused robust and immediate cell death in all neuronal lines, there was a significant recovery in membrane permeability at lower field strengths (Figure 5). This was also similar in cardiomyocytes in which membrane PI permeability recovered after 24 h (Figure 6). In contrast, cardiac fibroblast exhibited significantly higher cell death after 24 h, suggesting either the activation of a delayed cell death pathway or lack of recovery capacity.

IRE is now also commonly referred to as pulsed electric field (PEF) ablation and is seen as a minimally-thermal method that elicits its effect through the generation of nanopores in cell membranes [10,15]. The fundamental biophysics of the IRE is not yet fully elucidated, and this is an area of increasing basic science and clinical interest as it raises the potential of cell specific targeted ablation in complex tissues. This concept is of particular importance in cardiac ablation where a significant pitfall of thermal approaches is collateral tissue damage and in particular, phrenic nerve and oesophageal damage. Additionally, the targeting of the cardiac autonomic ganglionated plexi offers potential therapeutic advantages over more generalised cardiac ablations [16]. Moreover, it must be noted that the minimisation of the X-ray exposure in cardiac electrophysiology practices is vital for reducing the lifetime risk of cancer [17,18]. In this context, it is noted that recent work has established the possibility of using non-fluoro electroanatomical mapping systems for localising GPs [19].

Although the literature reports a series of lethal electroporation thresholds for different cell types [11,13,20,21], there is a lack of standardised pulse parameters and direct comparison within and across appropriate cell types. In the present study, we examined three neuronal lines of different origins that showed no significant difference in susceptibility to electroporation in comparison to cardiomyocytes. This clearly shows that for the given pulse parameters, in an in vitro suspension culture model, the lethal ablation electric field thresholds for neuronal and cardiac cells are comparable. The 1000 V/cm results in an effective injury for cardiomyocytes are in line with what have been reported by other in vitro studies [21]. In contrast, cardiac fibroblasts exhibited a lower susceptibility to damage in comparison to cardiomyocytes. This might be an important finding for clinical studies for cardiac ablation, as heterogeneity of cell types at target sites can impact the ablation size, depth, and success.

The data presented here showing the similar susceptibility of neurons and cardiac cells are in contrast with previous works. A recent study by Hunter et al. reported that rat ventricular cardiomyocytes had more susceptibility to damage for a given field strength in contrast to cortical neurons post electroporation [14]. The difference in our study is that we used neuronal lines that are more representative of the peripheral nervous system, suggesting a possible tissue selectivity to cell injury after electroporation. Furthermore, in our study, HL-1 cells were extracted from adult mouse atrial cardiomyocytes in comparison to neonatal rat ventricular myocytes (NRVM) used by others [14]. In addition, immature H9C2 cardiac cells in the Kaminska et al. study showed that electric field intensities of above 375 V/cm of five pulses of shorter 50 µs could cause 80% cell death [13]. The H9C2 myoblast cell line is an alternative model for cardiomyocytes but is immature; electrical activity and contractility are only observed in the presence of differentiation molecules, which is not typically conducted. Hence, this is very different to the HL-1 line used for this study, which are mature and have electrical functionality, making it a more suitable and appropriate cell line for this study.

The comparable ablation thresholds reported here might suggest the potential for damage in collateral structures in cardiac ablation. Phrenic nerve damage during a cardiac ablation procedure is a common complication with phrenic nerve palsy taking months to recover. Preclinical and clinical studies using electroporation have shown promising results with preservation of the phrenic nerve, however, these studies lack straightforward interpretation as different energy fields, and catheter electrode types and spacing are used [22,23,24]. An important factor to note is that in clinical applications, the axons of phrenic nerves are much more relevant rather than the mixture of cell bodies and axons studied in an in vitro scenario. This may significantly alter the IRE threshold and should be studied further.

At the same time, we cannot completely exclude the lack of other significant factors such as the in vivo three-dimensional geometrical positioning of the cells in the extracellular matrix as well as the electrical properties of the cells, which can vary nonlinearly and change through the duration of the electroporation [25]. Moreover, while in vitro tissue preparations and subsequent early clinical data suggest that IRE may exhibit relative tissue selectivity, it is unclear whether this arises from 3-dimensional tissue-cellular configuration (round or stretched), orientation (relative to the voltage gradient), cell type (oesophageal enterocytes/neurons/cardiomyocytes), and/or electroporation pulse sequences and waveform configuration. However, we can conclude from our model that cardiomyocytes have similar susceptibility to injury and damage as neuronal-like cells, and both are more vulnerable to cell death than cardiac fibroblasts.

In neuronal PC12 and SH-S5Y5 cell lines, at 60 pulses, the temporal dynamics of PI permeability shows that in general, a proportion of cells initially lose their membrane integrity and reseal after three hours, while a fraction of cells lose their integrity as a result of possibly a different cell death mechanism after 24 h. These results show that at higher electric fields and with 60 pulses, the percentage of cell death remains consistent with no significant alteration at 24 h in these neuronal lines. HL-1 cardiomyocytes exhibited some delayed cell death, while cardiac fibroblasts in all conditions showed the reverse effect with some initially viable cells, losing their membrane integrity after 24 h. This temporal effect is not consistent with previous in vivo pre-clinical and three-dimensional in vitro model studies [26,27], reinforcing the importance of consistent and detailed reporting of pulse parameters and variables that are found in the in vivo scenario. Our results highlight the similarity in ablation thresholds for cardiomyocytes and neurons in the cell suspension model and highlight the importance of monitoring cell death over time.

IRE offers a novel technique for cardiac ablation, but the pulse parameters require further optimisation. Considering cell type as a single variable, we showed that the ablation threshold in a cell suspension model is at least 1000 V/cm with 50 pulses of 100 μs duration for both neuronal-like cells and cardiomyocytes, and slightly higher for cardiac fibroblasts. Cell suspensions treated with IRE showed distinct temporal dynamics of cell death with no difference after 24 h between neuronal-like and cardiac cells. Future in vitro studies in modelling the same cells in confluent adhered layers and in three-dimensional constructs are necessary in order to more closely mimic the geometrical arrangement of cells during electroporation. It will also be beneficial to examine and compare the effect of these IRE thresholds with radiofrequency on different cell types in larger experimental designs. The safety of IRE and the possibility of collateral tissue damage has already been investigated pre-clinically [28,29,30], while early clinical data are also looking promising in this context [11,24]. While an advantage of the current approach is its simplicity and the direct comparison of a wide range of appropriate cell types, a limitation is that our in vitro model was performed in a suspension culture system and the results may therefore not be fully representative of tissue or organ structures. The neuronal lines used are a good representation of neurons, but better cell and tissue models need to be examined; a more relevant model close to neurons in ganglionated plexi or phrenic nerve is necessary. However, the combination of different neurons from different origins including peripheral neurons showed similar susceptibility and is unlikely to alter the degree of cell death.

## Figures and Tables

**Figure 1 jcm-10-05443-f001:**
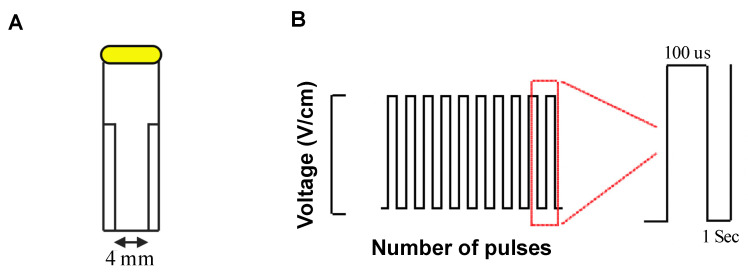
Irreversible electroporation experiment setup. Cell suspensions within a 4 mm gap cuvette placed in an external field (**A**) of monophasic pulses of different number and voltages in 1 Hz frequency as inter pulse interval and 100 us duration (**B**).

**Figure 2 jcm-10-05443-f002:**
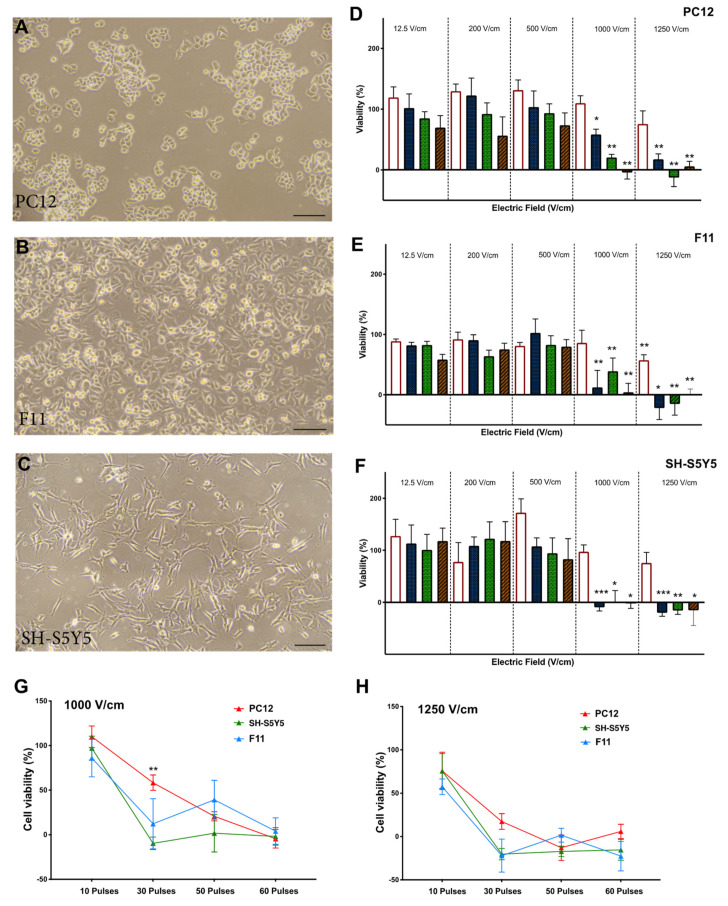
Neuronal viability is reduced by increasing electric field strength. The percentage of cell viability was normalised against control cells (no electroporation) in PC12 (**A**), F11 (**B**), and SH-S5Y5 (**C**) cells. (**D**) In PC12 suspension cells, there was a significant difference at 1000 and 1250 V/cm at 30, 50, and 60 pulses. (**E**) In F11 suspension cells, the cell viability was significantly reduced at 1000 V/cm of 30, 50, and 60 pulses and at 1250 V/cm at 10, 30, 50, and 60 pulses. (**F**) In SH-S5Y5 cells, at 30, 50, and 60 pulses cell viability significantly reduced. While the origin of these neuronal cell lines is different, no significant difference was observed at the threshold field strength of 1000 V/cm (**G**) and 1250 V/cm (**H**) among the neuronal lines with the exception of the difference between PC12 and SH-S5Y5 at 30 pulses. All data shown as mean ± SEM. Statistical significance performed using paired t-test and two-way ANOVA (* *p* < 0.05, ** *p* < 0.005, *** *p* < 0.001). Scale bar is 100 μm.

**Figure 3 jcm-10-05443-f003:**
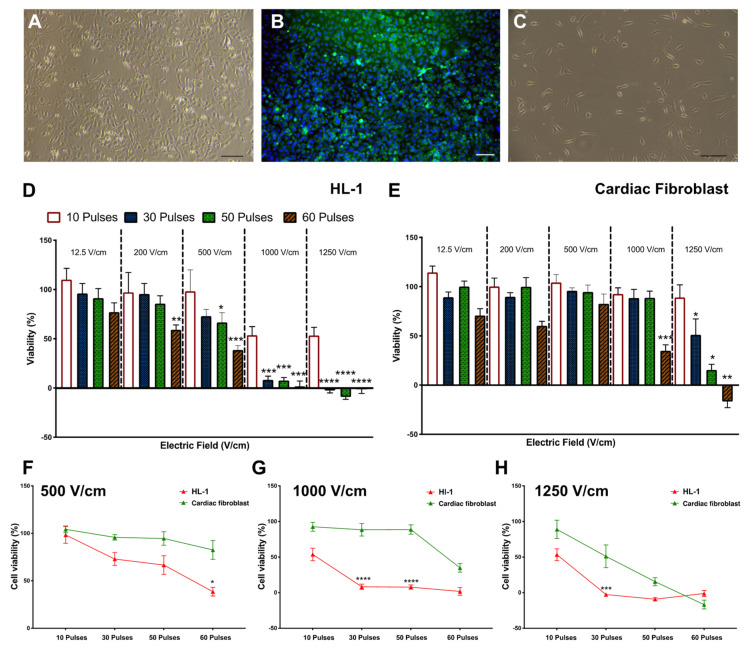
Cardiomyocyte and cardiac fibroblast viability are reduced by increasing electric field strength. (**A**) Representative brightfield image of HL-1 cells. (**B**) Fluorescent image of HL-1 cells stained for myosin (in green and nuclei counterstained with DAPI in blue). (**C**) Representative brightfield image of cardiac fibroblast. (**D**) HL-1 cells in suspension showed a significant reduction in cell viability at 60 pulses of all top voltages but substantially reduced to less than 10% viability at higher field strength of 1000 and 1250 V/cm. (**E**) Cardiac fibroblast cells in suspension only showed significant reduction at 1000 V/cm and 60 pulses with cell viability reaching less than 10% at 1250 V/cm and 60 pulses. The comparison of cardiomyocytes HL-1 cells and cardiac fibroblast shows that HL-1 cells have lower cell viability at 500 V/cm (**F**), 1000 V/cm (**G**), and 1250 V/cm (**H**). All data shown as mean ± SEM. Statistical significance performed using paired *t*-test and two-way ANOVA (* *p* < 0.05, ** *p* < 0.005, *** *p* < 0.001, **** *p* < 0.0001). Scale bar is 100 μm.

**Figure 4 jcm-10-05443-f004:**
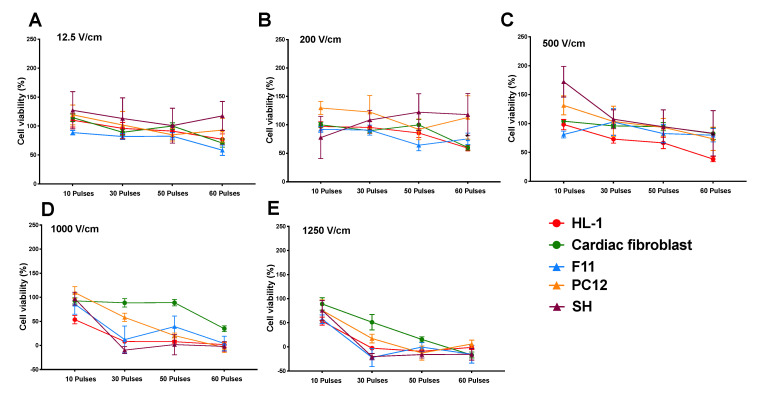
Influence of electric field parameters on cell viability of neurons and cardiac cells. Viability of cells after electroporation in different voltages of 12.5 V/cm (**A**), 200 V/cm (**B**), 500 V/cm (**C**), 1000 V/cm (**D**), and 1250 V/cm (**E**) showed no significant difference between neurons and HL-1 cardiomyocytes and cardiac fibroblast. All data shown as mean ± SEM.

**Figure 5 jcm-10-05443-f005:**
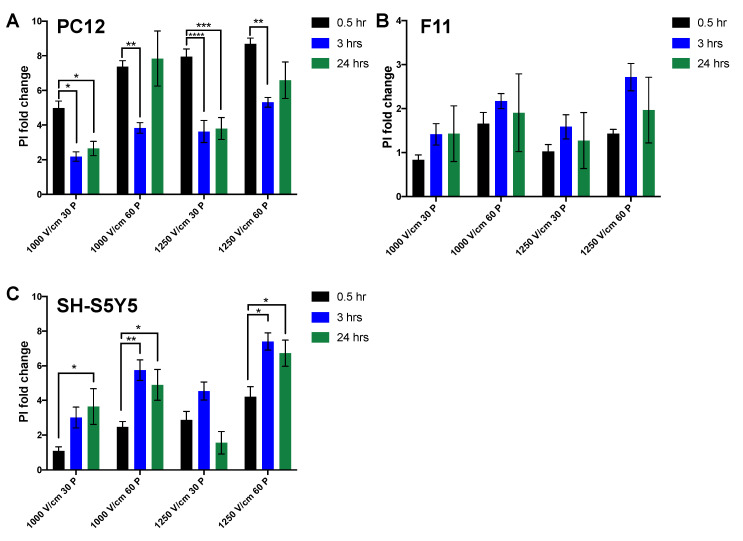
Temporal dynamics of neuronal cell death in response to electroporation. The evolution of membrane permeability to PI waw measured at 0.5 (in black), 3 (in blue), and 24 (in green) hours after treatment of cells in suspension with IRE with amplitudes of 1000, 1250 V/cm in 30 and 60 P (pulses) in PC12 (**A**), F11 (**B**), and SH-S5Y5 (**C**) cells. All data shown as mean ± SEM. Statistical significance performed using two-way ANOVA (* *p* < 0.05, ** *p* < 0.005, *** *p* < 0.001, **** *p* < 0.0001).

**Figure 6 jcm-10-05443-f006:**
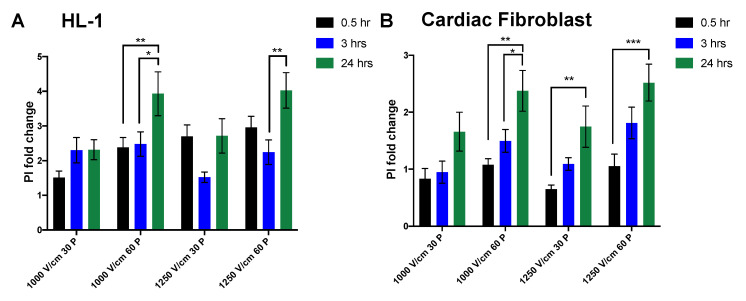
Temporal dynamics of cardiac cells in response to electroporation. The evolution of membrane permeability to PI were significantly decreased after 24 h at high voltage of 1250 V/cm in Hl-1 cardiomyocytes (**A**). In cardiac fibroblasts, in contrast, the PI^+^ fold change increased after 24 h (**B**). All data shown as mean ± SEM. Statistical significance performed using two-way ANOVA (* *p* < 0.05, ** *p* < 0.005, *** *p* < 0.001).

**Table 1 jcm-10-05443-t001:** Direct comparison of PI permeability among all neuronal lines at 0.5-h time point.

1000 V/Cm 30 Pulses	Mean ± SEM 1	Mean ± SEM 2	*p*-Value
F11 vs. PC12	0.84 ± 0.11	4.98 ± 0.41	<0.0001
F11 vs. SH-S5Y5	0.84 ± 0.11	1.08 ± 0.23	0.87 (ns)
PC12 vs. SH-S5Y5	4.98 ± 0.41	1.08 ± 0.23	<0.0001
**1000 V/cm 60 pulses**			
F11 vs. PC12	1.66 ± 0.25	7.37 ± 0.34	<0.0001
F11 vs. SH-S5Y5	1.66 ± 0.25	2.47 ± 0.32	0.16 (ns)
PC12 vs. SH-S5Y5	7.37 ± 0.34	2.47 ± 0.32	<0.0001
**1250 V/cm 30 pulses**			
F11 vs. PC12	1.03 ± 1.16	7.95 ± 0.44	<0.0001
F11 vs. SH-S5Y5	1.03 ± 1.16	2.88 ± 0.49	0.0010
PC12 vs. SH-S5Y5	7.95 ± 0.44	2.88 ± 0.49	<0.0001
**1250 V/cm 60 pulses**			
F11 vs. PC12	1.43 ± 0.10	8.69 ± 0.34	<0.0001
F11 vs. SH-S5Y5	1.43 ± 0.10	4.22 ± 0.57	<0.0001
PC12 vs. SH-S5Y5	8.69 ± 0.34	4.22 ± 0.57	<0.0001

Statistical significance performed using two-way ANOVA. ns = not significant.

**Table 2 jcm-10-05443-t002:** Direct comparison of PI permeability between Hl-1 cardiomyocytes and cardiac fibroblasts at the 0.5-h time point.

1000 V/cm 30 pulses	Mean ± SEM 1	Mean ± SEM 2	*p* Value
HL-1 vs. Cardiac fibroblast	1.51 ± 0.60	0.83 ± 0.44	0.70 [ns]
**1000 V/cm 60 pulses**			
HL-1 vs. Cardiac fibroblast	2.38 ± 0.89	1.08 ± 0.26	0.06 [ns]
**1250 V/cm 30 pulses**			
HL-1 vs. Cardiac fibroblast	2.70 ± 0.97	0.65 ± 0.18	0.0004
**1250 V/cm 60 pulses**			
HL-1 vs. Cardiac fibroblast	2.96 ± 0.95	1.05 ± 0.52	0.0014

Statistical significance performed using two-way ANOVA. ns = not significant.

## Data Availability

The data underlying this article will be shared on reasonable request to the corresponding author.

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
