# Peer review of "Establishing Irreversible Electroporation Electric Field Potential Threshold in A Suspension In Vitro Model for Cardiac and Neuronal Cells"

_jcm, 2021, doi:10.3390/jcm10225443_

Round 1
Reviewer 1 Report
Irreversible electroporation (IRE) is a promising nonthermal ablation technology for pulmonary vein isolation in patients with atrial fibrillation. Authors performed a comparison of different cell types within the cardiovascular system and identification of optimal voltage thresholds, that is mandatory for achieving a safe ablation modality. They used a simplified suspension culture electroporation model to explore IRE thresholds for cardiac applications. Cardiomyocyte and cardiac fibroblast viability are reduced by increasing electric field strength; however, cardiac fibroblasts exhibited a lower susceptibility to damage in comparison to cardiomyocytes. Results are nice as well as consistent, congratulations to authors. I also agree with authors that the main limitation is that in vitro model was performed in a suspension culture system and the results may therefore not be fully representative of tissue or organ structures
Comments: authors should more focus on the clinical perspective in both introduction and discussion section. In particular:
Introduction : Authors should better explain that, while mostly seen in elderly (Tini et al, doi: 10.1155/2020/2617970) , the atrial fibrillation can also affect young adults or adolescent. In particular, AF in the young may be precipitated by hypertension, hyperthyroidism (Frost et al doi:10.1001/archinte.164.15.1675), lifestyle factors such as endurance sport ( Mont et al https://doi.org/10.1093/europace/eun289), alcohol consumption, even smoking (Chamberlain et al, doi: 10.1016/j.hrthm.2011.03.038), and finally channelopathies (Vlachos et al, https://doi.org/10.1111/jce.14361; Platonov et al 10.1161/CIRCEP.119.007213; Mascia et al, doi: 10.1111/jce.13410). Considering all these factors is crucial and extremely important, since may prevent future AF attacks regardless of every ablation strategy (also considering electroporation), or may always re-evaluate its indication. Please cite this very important point in the introducton, including all suggested references.
Clinical benefits: In the discussion section on irreversible electroporation, I would also briefly include the importance of the existent “zero X-ray ablation approach” in electrophysiology as we definitely came in a new era (see: A New Era in Zero X-ray Ablation; Pani et al, doi: 10.1161/CIRCEP.117.005592; Yang et al, doi: 10.1016/j.amjcard.2016.08.014). In this scenario, authors should well explain that exposure related to X-ray transcatheter ablation carries small but non-negligible stochastic and deterministic effects on health (Sarkozy et al, doi: 10.1093/europace/eux252 ; Giaccardi et al. DOI: 10.1080/00015385.2020.1733303). Please cite these very important points, including all suggested references.
Limitations: at the same time in the discussion section authors should clarify that:
1) Although pulmonary veins may be isolated quickly and safely, we do not know if this translates into durable PV isolation with permanent lesions. Myocardial stunning may lead to temporary PV isolation, and insufficient electrode-tissue contact may result in shallow and discontinuous lesions with subsequent PV reconnection and AF recurrences.
2) Although author’s data suggest that understanding the optimal threshold of irreversible electroporation is vital for achieving a safe ablation modality without any side-effect in nearby cells, the efficacy of this new technique has to be studied in a larger trial, preferably compared with radiofrequency- or cryo-ablation in a randomized manner.
3) Possible damage to the brain, esophagus, or PVs has not been examined. Though animal experiments have demonstrated the safety of this new technique, future studies should also investigate possible collateral damage like thromboembolic brain damage, esophageal injury, or PV stenosis in patients. Please cyte: Wittkampf FH doi: 10.1111/j.1540-8167.2010.01863 , and Van Driel VJ et al doi:10.1016/j.hrthm.2015.05.012
Final comments: Technical results are definitely consistent .
Author Response
Reviewer 1
Reviewer comment: Irreversible electroporation (IRE) is a promising non-thermal ablation technology for pulmonary vein isolation in patients with atrial fibrillation. The authors performed a comparison of different cell types within the cardiovascular system and identification of optimal voltage thresholds, that is mandatory for achieving a safe ablation modality. They used a simplified suspension culture electroporation model to explore IRE thresholds for cardiac applications. Cardiomyocyte and cardiac fibroblast viability are reduced by increasing electric field strength; however, cardiac fibroblasts exhibited a lower susceptibility to damage in comparison to cardiomyocytes. Results are nice as well as consistent, congratulations to authors. I also agree with authors that the main limitation is that in vitro model was performed in a suspension culture system and the results may therefore not be fully representative of tissue or organ structures
Response: We are grateful for the reviewer’s comment and appreciation of our study. We totally understand the limitation of our suspension model system and in near future studies more representative model will be performed.
Comment 1: Introduction : Authors should better explain that, while mostly seen in elderly (Tini et al, doi: 10.1155/2020/2617970) , the atrial fibrillation can also affect young adults or adolescent. In particular, AF in the young may be precipitated by hypertension, hyperthyroidism (Frost et al doi:10.1001/archinte.164.15.1675), lifestyle factors such as endurance sport ( Mont et al https://doi.org/10.1093/europace/eun289), alcohol consumption, even smoking (Chamberlain et al, doi: 10.1016/j.hrthm.2011.03.038), and finally channelopathies (Vlachos et al, https://doi.org/10.1111/jce.14361; Platonov et al 10.1161/CIRCEP.119.007213; Mascia et al, doi: 10.1111/jce.13410). Considering all these factors is crucial and extremely important, since may prevent future AF attacks regardless of every ablation strategy (also considering electroporation), or may always re-evaluate its indication. Please cite this very important point in the introducton, including all suggested references.
Response: We thank the reviewer for this comment. We have now included this in the introduction (page 4) of the manuscript with the suggested references " AF is the most common form of cardiac arrhythmia widely associated with increased age though also occurs in young adults and adolescent (1). In younger adults, AF is usually precipitated by many underlying factors such as hypertension, hyperthyroidism (2), alcohol consumption, smoking (3) and channelopathies (4)”.
Comment 2: Clinical benefits: In the discussion section on irreversible electroporation, I would also briefly include the importance of the existent “zero X-ray ablation approach” in electrophysiology as we definitely came in a new era (see: A New Era in Zero X-ray Ablation; Pani et al, doi:
Response: We appreciate the reviewer’s comment. We have now included this in our discussion (Page 12) of the manuscript with the suggested references “Moreover, it must be noted that the minimization of the X-ray exposure in cardiac electrophysiology practices is vital for reducing the lifetime risk of cancer (17,18). In this context it is noted that recent work has established the possibility of using non-fluoro electroanatomical mapping systems for localizing GPs (19)”.
Comment 3: Limitations: at the same time in the discussion section authors should clarify that:
1) Although pulmonary veins may be isolated quickly and safely, we do not know if this translates into durable PV isolation with permanent lesions. Myocardial stunning may lead to temporary PV isolation, and insufficient electrode-tissue contact may result in shallow and discontinuous lesions with subsequent PV reconnection and AF recurrences.
2) Although author’s data suggest that understanding the optimal threshold of irreversible electroporation is vital for achieving a safe ablation modality without any side-effect in nearby cells, the efficacy of this new technique has to be studied in a larger trial, preferably compared with radiofrequency- or cryo-ablation in a randomized manner.
3) Possible damage to the brain, esophagus, or PVs has not been examined. Though animal experiments have demonstrated the safety of this new technique, future studies should also investigate possible collateral damage like thromboembolic brain damage, esophageal injury, or PV stenosis in patients. Please cyte: Wittkampf FH doi:
Response: We appreciate reviewer’s comment for the limitation parts of our study. We have now included these parts in the end of the discussion (Page 16) of the manuscript “It will be also beneficial to examine and compare the effect of these IRE thresholds with radiofrequency on different cell types in larger experimental designs. The safety of IRE and the possibility of collateral tissue damage has already been investigated pre-clinically (28–30) while early clinical data is also looking promising in this context (11, 24)”.
Final comments: Technical results are definitely consistent.

Reviewer 2 Report
Summary: In this article, the authors conduct an in vitro investigation of cell viability following IRE treatment in 3 rodent, malignant neuronal cell lines (PC12, F11, SH-S5Y5), 1 rodent atrial cardiomyocyte cell line (HL-1), and 1 human cardiac fibroblast cell line. The premise of this manuscript is to attain in vitro cell death profiles from which a direct comparison can be made across cell lines present during cardiac ablation of arrhythmic tissue. Most data is collected for cells in which a direct comparison cannot be made (human vs rodent cells, malignant neuronal cells vs. healthy tissue). Prior to resubmission, the reviewer recommends the dataset be expanded to include both rodent and human cell lines of healthy neurons, cardiomyocytes, and cardiac fibroblasts.
Author Response
Reviewer 2
Summary: In this article, the authors conduct an in vitro investigation of cell viability following IRE treatment in 3 rodent, malignant neuronal cell lines (PC12, F11, SH-S5Y5), 1 rodent atrial cardiomyocyte cell line (HL-1), and 1 human cardiac fibroblast cell line. The premise of this manuscript is to attain in vitro cell death profiles from which a direct comparison can be made across cell lines present during cardiac ablation of arrhythmic tissue. Most data is collected for cells in which a direct comparison cannot be made (human vs rodent cells, malignant neuronal cells vs. healthy tissue). Prior to resubmission, the reviewer recommends the dataset be expanded to include both rodent and human cell lines of healthy neurons, cardiomyocytes, and cardiac fibroblasts.
Response: We agree with the reviewers comment in the sense that in the ideal experimental model the cells used should represent the target tissue as closely as possible. In this study the use of human cardiomyocytes and human autonomic neurons were not available to the research team. We choose well validated and representative models as an alternative. The use of human iPSCs are considered the best model for this type of in-vitro study, however, access to iPSC banks and their differentiation to desired cell types are hugely expensive and time demanding. PC12, F11, SH-S5Y5 have been widely used in many in-vitro modelling. For instance, SH-S5Y5 have been extensively used to model for dopaminergic neurons in Parkinson research (Xicoy et al. 2017; Pan et al. 2020; Krishna et al. 2014; Borland et al. 2008; Xie et al. 2010), F11 in in vitro model system of peripheral sensory neurons (Yin et al. 2016; Haberberger et al. 2020; Wood et al. 1990; Ambrosino et al. 2019), and PC12 cells in in vitro model of neurotoxicity, neuroprotection, neurosecretion, neuroinflammation and synaptogenesis (Wiatrak et al. 2020), with their extensive use in neurobiology as they exhibit mature dopaminergic neuron (Wang et al. 2015; Malagelada and Greene 2008) and sympathetic (Grau and Greene 2012) neuronal like phenotypes. We are confident that the work presented represent a novel and important contribution for high throughput testing of ablation thresholds in relevant cell models for cardiac studies.
Bibliography
Ambrosino, P., Soldovieri, M.V., Di Zazzo, E., et al. 2019. Activation of Kv7 Potassium Channels Inhibits Intracellular Ca2+ Increases Triggered By TRPV1-Mediated Pain-Inducing Stimuli in F11 Immortalized Sensory Neurons. International Journal of Molecular Sciences 20(18).
Borland, M.K., Trimmer, P.A., Rubinstein, J.D., et al. 2008. Chronic, low-dose rotenone reproduces Lewy neurites found in early stages of Parkinson’s disease, reduces mitochondrial movement and slowly kills differentiated SH-SY5Y neural cells. Molecular Neurodegeneration 3, p. 21.
Grau, C.M. and Greene, L.A. 2012. Use of PC12 cells and rat superior cervical ganglion sympathetic neurons as models for neuroprotective assays relevant to Parkinson’s disease. Neurotrophic Factors.
Haberberger, R.V., Barry, C. and Matusica, D. 2020. Immortalized dorsal root ganglion neuron cell lines. Frontiers in Cellular Neuroscience 14, p. 184.
Krishna, A., Biryukov, M., Trefois, C., et al. 2014. Systems genomics evaluation of the SH-SY5Y neuroblastoma cell line as a model for Parkinson’s disease. BMC Genomics 15, p. 1154.
Malagelada, C. and Greene, L.A. 2008. PC12 cells as a model for Parkinson’s disease research. Parkinson’s Disease.
Pan, X., Liu, X., Zhao, H., Wu, B. and Liu, G. 2020. Antioxidant, anti-inflammatory and neuroprotective effect of kaempferol on rotenone-induced Parkinson’s disease model of rats and SH-S5Y5 cells by …. Journal of Functional Foods.
Wang, W.L., Dai, R., Yan, H.W. and Han, C.N. 2015. Current situation of PC12 cell use in neuronal injury study. of Biotechnology for.
Wiatrak, B., Kubis-Kubiak, A., Piwowar, A. and Barg, E. 2020. PC12 cell line: cell types, coating of culture vessels, differentiation and other culture conditions. Cells 9(4).
Wood, J.N., Bevan, S.J., Coote, P.R., et al. 1990. Novel cell lines display properties of nociceptive sensory neurons. Proceedings. Biological Sciences / the Royal Society 241(1302), pp. 187–194.
Xicoy, H., Wieringa, B. and Martens, G.J.M. 2017. The SH-SY5Y cell line in Parkinson’s disease research: a systematic review. Molecular Neurodegeneration 12(1), p. 10.
Xie, H., Hu, L. and Li, G. 2010. SH-SY5Y human neuroblastoma cell line: in vitrocell model of dopaminergic neurons in Parkinson’s disease. Chinese medical journal.
Yin, K., Baillie, G.J. and Vetter, I. 2016. Neuronal cell lines as model dorsal root ganglion neurons: A transcriptomic comparison. Molecular pain.

Round 2
Reviewer 1 Report
Manuscript significantly improved with important clinical messages
Reviewer 2 Report
Accept in present form.